# Comprehensive, comparative evaluation of 25 automated SARS-CoV-2 serology assays

Wayne Dimech,[1] Shannon Curley,[1] Jing Jing Cai[1]

ABSTRACT    The onset of the COVID-19 pandemic resulted in hundreds of *in vitro* devices coming to market, facilitated by regulatory authorities allowing "Emergency Use," without a prior comprehensive evaluation of performance. The World Health Organization released Target Product Profiles specifying acceptable performance characteristics for SARS-CoV-2 devices. We evaluated 16 automated serology test kits that detect IgG or Total antibodies to SARS-CoV-2, along with a further nine tests that detect IgM-specific antibodies. All IgG or Total antibody tests reported a concordance with recent infection at 83.9% or greater, with 11/16 tests having greater than 90% sensitivity. All 16 tests reported greater than 96.3% specificity. There was a low level of false reactivity when testing all 25 tests on panels of samples containing potentially cross-reacting or interfering substances. A range of results were reported when testing seroconversion and dilution panels. The study further demonstrates that a comprehensive evaluation of the performance of test kits assessed against defined specifications is essential for the selection of test kits, especially in a pandemic setting.

IMPORTANCE    We have previously highlighted the fact that hundreds of SARS-CoV-2 serology tests were released months after the onset of the COVID-19 pandemic. Of the hundreds of studies investigating the test kits' performance, few were comparative reports, using the same comprehensive sample set across multiple tests. Recently, we reported a comparative assessment of 35 rapid diagnostic tests (RDTs) or microtiter plate enzyme immunoassays (EIA) for use in low- and middle-income countries, using a large sample set from individuals with a history of COVID-19. Only a few tests meet WHO Target Product Profile performance requirements. This study reports on the performance of a further 25 automated SARS-CoV-2 immunoassays using the same panel of samples. The results highlight the better analytical and clinical performance of automated serology test kits compared with RDTs, and the importance of independent comparative assessments to inform the use and procurement of these tests for both diagnostic and epidemiological investigations.

KEYWORDS    anti-SARS-CoV-2 antibodies, serology, evaluation, automated assays

I n November 2019, a novel acute respiratory disease (COVID-19) caused by a new coronavirus (SARS-CoV-2) was first recognized. Some of the first commercial diagnostic tests available on the market were rapid, serology test kits (1). The emergence of automated tests for anti-SARS-CoV-2 was on average several months later, with the majority becoming available after the second half of 2020 and beyond. Test kit manufacturers were able to provide the tests under Emergency Use Listing (EUL), where regulators required limited clinical and analytical performance information (2, 3). This allowed rapid access to diagnostics but removed the stringent regulatory requirements applied to other similar assays. Many studies reviewing the performance of these test kits were published (4–13), but few studies used a comprehensive panel

Editor Oliver Laeyendecker, National Institute of Allergy and Infectious Diseases, Baltimore, Maryland, USA

Ad Hoc Peer Reviewer Adolfo Firpo-Betancourt, Icahn School of Medicine at Mount Sinai, New York, New York, USA

Address correspondence to Wayne Dimech, wayne@nrlquality.org.au.

The authors declare no conflict of interest.

of well-characterized samples to evaluate the performance in a manner that would normally be required by regulators (1, 14–16).

The National Serology Reference Laboratory, Australia (NRL), in collaboration with the World Health Organization (WHO), implemented a study to assess 34 RDT and EIA using the well-characterized panel of samples (17). In addition to this WHO study, NRL offered a similar evaluation service to other manufacturers of laboratory-based, automated SARS-CoV-2 serology tests, drawing from the same panels of samples. A total of 16 IgG or Total antibody test kits were included in the study. These test kits detected either IgG only or Total antibodies, against viral spike or nucleocapsid antigens. A further nine test kits that detected anti-SARS-CoV-2 IgM were included in the study. Individual test kit summary statistics results of the study were published on the NRL website after each evaluation. This report represents a comparison of the performance of the tests.

## RESULTS

The results of testing for each performance criterion are detailed below.

### Concordance with recent infection

All 16 test kits testing for IgG or Total antibodies reported a concordance with recent infection of greater than 83.9%, with five test kits reporting less than 90% sensitivity (Fig. 1). The Ortho VITROS Total was the only test kit with 100% sensitivity.

### Clinical specificity

All 16 test kits detecting IgG or Total antibodies reported a specificity of greater than 95%, with VIRCLIA IgG having the lowest at 96.3% (Fig. 1). Euroimmun Spike IgG, Sysmex HISCL N IgG, and Roche Elecsys N Total all reported 100%.

### Analytical sensitivity/lot-to-lot variation

The test kits detecting IgG or Total antibodies reported reactivity when tested on three dilution series, ranging from a dilution of 1:2 to 1:1,024, indicating differences in analytical sensitivity between test kits, as well as between samples. No test kit reported a difference of greater than one-fold dilution difference between the two test kit lot numbers tested (Fig. 2).

### Cross-reacting and interfering substances

All test kits, excluding the bioMerieux VIDAS IgM (due to insufficient tests) were tested against the 55-member cross-reacting panel and 35-member interfering substance panel (Table 1). Euroimmun NCP IgG (CMV IgM, *Chlamydia psittaci* IgM and Influenza A positive) and VIRCLIA IgG (CMV IgM, HIV and Parainfluenza positive) reported 3/55 cross-reacting samples and Sysmex HISCL S-IgG (two rheumatoid factors positive and one icteric sample) reported 3/35 interfering substance samples as reactive. All other test kits reported two or fewer reactive results for both the cross-reacting and interfering substance samples. Eight test kits (DiaSorin LIAISON Tri-S IgG, Ortho VITROS IgG, Ortho VITROS Total, Roche Elecsys S, Siemens Atellica Total, Sysmex HISCL N-IgG, Sysmex HISCL N-IgM, and Sysmex HISCL S-IgM) reported no cross-reacting or interfering reactivity.

### Seroconversion panels

All 25 test kits were tested in the five seroconversion panels. Generally, IgG and IgM reactivity was detectable at approximately the same bleed for each individual. All IgM test kits demonstrated seroconversion followed by reversion to negativity on one or more of the panel samples (Fig. S1). All of the IgG and Total antibody tests remained reactive to the last bleed, the exception being Siemens Atellica IgG which reported negative results for two of the last three bleeds of patent MRNCOV-512. This may be false negative reactivity. Of note, Sysmex HISCL N-IgM only reported that 3/60 samples

| Abbreviation | Concordance with recent infection [95% CI range] | | Specificity [95% CI range] | |
|---|---|---|---|---|
| | IgG | Total Ab | IgG | Total Ab |
| Abbott Alinity IgG II Quant | 99.5 [96.8 – 100] | NA | 98.0 [95.5 – 99.2] | NA |
| Abott Alinity IgG | 94.5 [90.1 – 97.1] | NA | 99.3 [97.4 – 99.9] | NA |
| Abbott Architect IgG | 93.0 [88.2 – 96.0] | NA | 98.7 [96.4 – 99.6] | NA |
| Abbott Architect IgG II Quant | 99.0 [96.0 – 99.8] | NA | 97.3 [94.6 – 98.8] | NA |
| DiaSorin Liason Tri-S IgG | 94.5 [90.1 – 97.1] | NA | 99.7 [97.9 - 100] | NA |
| Euroimmune Spike  IgG | 89.4 [84.1 – 93.2] | NA | 100 [98.4 – 100] | NA |
| Euroimmune NCP IgG | 87.9 [82.4 – 92.0] | NA | 99.0 [96.9 – 99.7] | NA |
| Sysmex HISCL N IgG | 96.5 [92.6 – 98.5] | NA | 100 [98.4 - 100] | NA |
| Sysmex HISCL S IgG | 89.9 [84.7 – 93.6] | NA | 99.7 [97.9 - 100] | NA |
| Ortho VITROS IgG | 88.9 [83.5 - 92.8] | NA | 99.3 [97.3 - 99.9] | NA |
| Ortho VITROS Total | NA | 100 [97.6 - 100] | NA | 99.0 [96.8 - 99.7] |
| Roche Elecsys N Total | NA | 97.5 [93.9 – 99.1] | NA | 100 [98.4 - 100] |
| Roche Elecsys S Total | NA | 99.5 [96.8 - 100] | NA | 99.3 [97.3 – 99.9] |
| Siemens Atellica IgG Quant | 83.9 [77.9 – 88.6] | NA | 97.3 [94.6 – 98.8] | NA |
| Siemens Atellica Total Quant | 99.5 [96.8 – 100] | NA | 99.7 [97.9 – 100] | NA |
| VIRCLIA IgG | 95.0 [90.7 – 97.4] | NA | 96.3 [93.3 – 98.1] | NA |

FIG 1  Concordances with recent infection ($n$ = 199), and specificities ($n$ = 300) of automated SARS-CoV-2 serology testing for either IgG or Total antibodies. Concordance and specificity results are presented as a heat map, with shades of green representing results of >90%, and shades of yellow representing results of between 60% and 90%.

are reactive. This was the only nucleocapsid IgM test kit possibly indicating that an IgM response to that antigen was not elicited in the five individuals.

**FIG 2** Results of testing two lots (A and B) of automated SARS-CoV-2 IgG or Total antibody test kits on three serial dilutions of antibody-positive samples, with R indicating reactive results and NR indicating non-reactive results.

## Sero-reversion panels

The nine test kits detecting IgM were tested on the sero-reversion panels. A range of reactivities was reported by each test kit, ranging from all samples within a panel being reactive to all being non-reactive (Fig. S2). At least 4/9 test kits reported at least one reactive result for each of the 10 panels.

## Repeatability

Only 3/16 test kits had a percent coefficient of variation (%CV) of greater than 5% (Table 2). The two microtitre plate EIAs had the greatest imprecision—Euroimmun NCP IgG (23.0%) and Euroimmun Spike IgG (25.6%). The two Roche tests reported a %CV of less than 1.0%.

## DISCUSSION

Serology tests for SARS-CoV-2 became available early in 2020, mainly in the form of RDTs (2). The diagnostic utility of these tests was unknown at the time, but many jurisdictions allowed their use as EUL (1, 18, 19). The advent of automated serology testing followed, with most major suppliers of continual access testing platforms developing and releasing serology test kits. Number of studies sought to assess the performance of these tests (6, 10, 13, 20, 21). In a recently published evaluation in collaboration with WHO, we described the performance of RDTs and EIA used to detect antibodies to SARS-CoV-2 (17). The majority of the 34 tests evaluated failed to reach the WHO Target Product Profiles (TPP) (22), with sensitivity and specificity ranging from 60.1% to 100% and 56.0% to 100% respectively.

In contrast, the automated assays evaluated using the same panel of samples reported superior performance characteristics compared with the RDT and EIAs. Whereas the TPP performance characteristics of RDT required sensitivity to being acceptable at $\geq$90% and desirable at $\geq$95% and specificity being acceptable at $\geq$97% and desirable at $\geq$99%; higher throughput assays had acceptable and desirable sensitivities of $\geq$95% and $\geq$98% and specificities of $\geq$97% and $\geq$99%, respectively (22). Eight and 15 of the 16

**TABLE 1** Cross-reacting panel comprised 55 samples containing common cross-reacting analytes and 35 samples containing potentially interfering substances

| Analyte | Number of samples |
|---|---|
| CMV IgM positive | 4 |
| EBV VCA IgM positive | 2 |
| Influenza A positive | 3 |
| Influenza B positive | 3 |
| Hepatitis A IgM positive | 1 |
| Hepatitis B e-antigen positive | 3 |
| Hepatitis B surface antigen | 5 |
| Hepatitis B surface antigen/Hepatitis B c IgM positive | 1 |
| Hepatitis B surface antigen/Hepatitis B c IgM/Hepatitis B-e antigen positive | 1 |
| Hepatitis C virus antibody positive | 4 |
| HIV antibody positive | 8 |
| Malaria antibody positive | 5 |
| Mycoplasma IgM positive | 1 |
| Parainfluenza positive | 1 |
| Parvovirus B19 IgM positive | 2 |
| Psittacosis IgM positive | 1 |
| Rubella IgM positive | 1 |
| Syphilis positive | 6 |
| Toxoplasma IgM positive | 3 |
| Icteric | 5 |
| Hemolyzed | 5 |
| High bilirubin | 7 |
| High lipid | 5 |
| Anti-nuclear antibody positive | 5 |
| Double-stranded DNA (Lupus) antibody positive | 3 |
| Rheumatoid factor positive | 5 |

automated test kits testing for IgG or Total antibodies for SARS-CoV-2 achieved desirable sensitivity and specificity performance, respectively.

Only five and six RDT or EIA test kits reported no false reactivity when tested on the 55 samples containing potentially cross-reacting substances and the 35 samples containing potentially interfering substances, respectively. One RDT reported 47/55 and 26/35 reactive results for the cross-reacting and interfering substance panels. Compared with the 35 RDT and EIA test kits previously evaluated, the automated tests reported fewer false reactive results when tested on samples containing potentially interfering substances, with no automated test reporting more than three false reactive results on either cross-reacting or interfering panel samples. Eight tests reported no false reactive results on either panel.

Testing the automated tests on seroconversion panels and dilution series demonstrated a range of analytical sensitivities for IgG, IgM, and Total antibody tests. This information may be useful in determining potential uses for serology assays, although it is well-accepted that serology is not useful in the clinical diagnosis of COVID-19 (23). Understanding the rise and fall of antibodies post-infection may be useful in understanding if a person has been recently infected with SARS-CoV-2 even if rapid antigen or RNA tests are negative. The findings also contribute to the understanding that tests perform differently and that a scientifically robust assessment of their performance is vital for their selection and use.

The onset of the COVID-19 pandemic has highlighted some potential deficiencies in the way the scientific and regulatory communities react to such health emergencies. While the use for EUL of *in vitro* diagnostics devices (IVD) served the purpose of allowing rapid access to these tools by health workers, the decision also allowed many inferior test kits onto the market. EUL generally allowed manufacturer-declared evidence, without

TABLE 2 Repeatability results, expressed as %CV, of automated SARS-CoV-2 serology test reporting quantitative or signal-to-cut-off results

| Test abbreviation | Repeatability (%CV) |
| --- | --- |
| Roche Elecsys N Total | 0.8 |
| Roche Elecsys S | 0.9 |
| DiaSorin LIAISON Tri-S IgG | 1.2 |
| Abbott Alinity IgG | 1.6 |
| Ortho VITROS IgG | 1.6 |
| Abbott Architect IgG | 1.7 |
| Ortho VITROS Total | 1.9 |
| Sysmex HISCL S-IgG | 1.9 |
| Abbott Architect IgG II Quant | 2.5 |
| Siemens Atellica Total | 2.6 |
| Sysmex HISCL N-IgG | 2.7 |
| Abbott Alinity IgG II Quant | 2.9 |
| Siemens Atellica IgG | 4.6 |
| VIRCLIA IgG | 9.8 |
| Euroimmun NCP IgG | 23.0 |
| Euroimmun Spike IgG | 25.6 |

comprehensive data to support the claims (23). It took several years for regulatory authorities to re-impose stringent regulatory requirements onto manufacturers. At the same time, numerous studies reported the performance characteristics of these test kits. Most of these studies used low-volume remnants of clinical samples, poorly designed protocols, and questionable conclusions, resulting in conflicting assessments of performance. Many studies were published without peer review (14). A limitation of this study is that it used commercially acquired from non-hospitalized patients from the USA or Germany in the panel of positive samples, acquired early in the pandemic. Therefore, the SARS-CoV-2 antibodies detected were post-infection with the Wuhan strain. The ability of these assays to detect antibodies arising from infections with other variants of concern or post-immunization was not assessed.

It is important that lessons are learned from this situation as outlined in a recent paper by FIND (2). The evaluation of the performance of test kits is a significant undertaking that requires well-developed protocols, panels of well-characterized samples, and thoughtful analysis and reporting of results. The 100 days mission report states that "Stringent Regulatory Authorities" should work together to define international assessment protocols and develop guiding principles, alongside more effective quality assurance processes (24). WHO has a network of IVD prequalification evaluation laboratories that perform this testing, as well as several other expert laboratories such as the Paul Ehrlich Institute and others. We would strongly recommend that a network of laboratories such as these be strengthened to respond rapidly to future health emergencies such as the original SARS, MERS, Zika, COVID-19, Ebola, MPox, and other outbreaks that continually and increasingly arise. Access to clinical samples, ethics, material transfer agreements, importation permits, templated protocols, and other infrastructure required to evaluate novel IVDs in an emergency setting will be vital to advise the government, regulators and health workers on the performance of these IVDs.

## MATERIALS AND METHODS

### Test kit selection

A detailed study protocol was developed, and samples used for testing were acquired. Each test kit manufacturer was provided the study protocol and was invited to participate in the evaluation. All manufacturers provided test kits and associated reagents to NRL at no charge. No exclusion criteria were implemented, however, test kits used to

**TABLE 3** Final list of test kits included in the WHO SARS-CoV-2 serology evaluations, including abbreviations used in the report

| Abbreviation | Manufacturer | Product name | Product code | IFU[f] version | Test type | Antibody class | Antigen |
|---|---|---|---|---|---|---|---|
| Abbott Alinity IgG II Quant | Abbott Diagnostics | Alinity i SARS-CoV-2 IgG II Quant | 06S61 | April 2021 | CMIA[c] | IgG (quantitative) | Spike (RBD)[d] |
| Abbott Alinity IgM | Abbott Diagnostics | Alinity i SARS-CoV-2 IgM | 06R91 | August 2020 | CMIA | IgM | Spike |
| Abbott Alinity IgG | Abbott Diagnostics | Alinity SARS-CoV-2 IgG | 06R90 | June 2020 | CMIA | IgG | Nucleocapsid |
| Abbott Architect IgG | Abbott Diagnostics | Architect SARS-CoV-2 IgG | 6R86 | April 2020 | CMIA | IgG | Nucleocapsid |
| Abbott Architect IgG II Quant | Abbott Diagnostics | SARS-CoV-2 IgG II Quant | 6S60 | February 2021 | CMIA | IgG (quantitative) | Spike (RBD) |
| Abbott Architect IgM | Abbott Diagnostics | Architect SARS-CoV-2 IgM | 6R87 | August 2020 | CMIA | IgM | Spike |
| bioMerieux VIDAS IgM | bioMerieux | VIDAS SARS-COV-2 IgM (9COM) | 423833 | 055963 - 02 - 2020-06 - en | | IgM | Unspecified recombinant SARS-CoV-2 antigen |
| DiaSorin Liaison Tri-S IgG | DiaSorin | LIAISON SARS-CoV-2 Trimeric S IgG | 311510 | EN - 54286 - 2021-02 & EN - 54529 - 2021-04 | CLIA | IgG (quantitative) | Spike |
| DiaSorin Liaison IgM | DiaSorin | LIAISON SARS-CoV-2 IgM | 311470 | 200/008-030, 04 - 2021-04 | CLIA[a] | IgM | Spike RBD antigen (mammalian cells) |
| Euroimmune Spike IgG | Euroimmun | Anti-SARS-CoV-2 ELISA (IgG) | EI 2606-9601 G | EI_2606G_A_UK_C09.docx (2020-10-14) | EIA | IgG | Spike recombinant S1 domain (human cell line HEK 293) |
| Euroimmune NCP IgG | Euroimmun | Anti-SARS-CoV-2 NCP ELISA (IgG) | EI 2606-9601-2 G | EI_2606-2G_A_UK_C04.docx (2020-07-17) | EIA | IgG | Modified nucleocapsid protein |
| Sysmex HISCL N IgG | Sysmex Corporation | HISCL SARS-CoV-2 N-IgG | RUO | December 2020 | CLEIA[b] | IgG | Nucleocapsid |
| Sysmex HISCL S IgG | Sysmex Corporation | HISCL SARS-CoV-2 S-IgG | AP682349 | BT378430 (06/2021) | CLEIA | IgG | Spike |
| Sysmex HISCL N IgM | Sysmex Corporation | HISCL SARS-CoV-2 N-IgM | RUO | December 2020 | CLEIA | IgM | Nucleocapsid |
| Sysmex HISCL S IgM | Sysmex Corporation | HISCL SARS-CoV-2 S-IgM | RUO | December 2020 | CLEIA | IgM | Spike |
| Ortho VITROS IgG | Ortho-Clinical Diagnostics, Inc. | VITROS Immunodiagnostic Products Anti-SARS-CoV-2 IgG | 619 9919 | GEM1292_US_EN (Version 4.2) | CLIA | IgG | Spike |
| Ortho VITROS Total | Ortho-Clinical Diagnostics, Inc. | VITROS Immunodiagnostic Products Anti-SARS-CoV-2 Total | 619 9922 | GEM1293_US_EN (Version 3.2) | CLIA | Total antibody (including IgM/IgG/IgA) | Spike (S1) |
| Roche Elecsys N Total | Roche Diagnostics GmbH | Elecsys Anti-SARS-CoV-2 | 09203079190 | 09203079500 Ver: 4.0 (03/2021) | ECLIA[e] | Total antibody (including IgG) | Recombinant nucleocapsid |
| Roche Elecsys S Total | Roche Diagnostics GmbH | Elecsys Anti-SARS-CoV-2 S | 09289275190 | 09289275500 Ver: 2.0 (05/2021) | ECLIA | Total antibody (including IgG) | Spike (RBD) |
| Siemens Atellica IgG Quant | Siemens Healthineers | Atellica IM SARS-CoV-2 IgG (sCOVG) | 11207386 | 11208937_EN Rev. 01, 2021-01 | CLIA | IgG (quantitative) | Spike (S1 RBD) |
| Siemens Atellica Total Quant | Siemens Healthineers | Atellica IM SARS-CoV-2 Total (COV2T) | 11206711 | 11208933_EN Rev. 03, 2021-06 | CLIA | Total antibody (IgG and IgM) (quantitative) | Spike (S1 RBD) |

*(Continued on next page)*

**TABLE 3** Final list of test kits included in the WHO SARS-CoV-2 serology evaluations, including abbreviations used in the report (*Continued*)

| Abbreviation | Manufacturer | Product name | Product code | IFU[f] version | Test type | Antibody class | Antigen |
|---|---|---|---|---|---|---|---|
| VIRCLIA IgG | Vircell, S.L. | COVID-19 VIRCLIA IgG MONOTEST | VCM097 | L-VCM097-EN-04 (2020-11-16) | CLIA | IgG | Spike and nucleocapsid |
| VirClia IgM+IgA | Vircell, S.L. | COVID-19 VIRCLIA IgM+IgA MONOTEST | VCM098 | L-VCM098-EN-04 (2020-11-16) | CLIA | IgM & IgA | Spike and nucleocapsid |

[a]CLIA, chemiluminescence immunoassay.
[b]CLEIA, chemiluminescent enzyme immunoassay.
[c]CMIA, chemiluminescent microparticle immunoassay.
[d]RBD, receptor-binding domain.
[e]ECLIA, electrochemiluminescence immunoassay.
[f]IFU, instructions for use.

solely detect IgM-specific SARS-CoV-2 were tested on a limited set of panels. In total, 25 test kits from nine manufacturers were included in the study (Table 3). Not all tests were commercially available, with some being research use only.

## Sample panels

The panels of samples used in the study are presented in detail elsewhere (17). Briefly, the test kits were evaluated using the following panels.

### Sensitivity/concordance with recently confirmed SARS-CoV-2 infection

A total of 199 commercially acquired plasma samples were obtained from non-hospital-ized individuals with a recent history of clinical infection with ancestral SARS-CoV-2, confirmed by various commercial NATs. All samples were collected between January and April 2020, and between 14 and 71 days post-infection.

### Specificity

A total of 300 plasma samples obtained from health blood donors were stored in NRL's sample bank, having been collected prior to November 2019. These samples were assumed to be negative for SARS-CoV-2 and no additional confirmatory testing was performed.

### Analytical sensitivity/lot-to-lot variation

The sensitivity panel samples consisted of 10 doubling dilutions of three of the sensitivity panel samples, from 1:2 to 1:1,024, prepared in human plasma negative for SARS-CoV-2 antibodies. All dilutions were tested on two reagent lots.

### Cross-reacting and interfering substances

A total of 55 plasma or serum samples were known to contain potentially cross-reacting analytes, and a further 35 serum or plasma samples containing potentially interfering substances were tested in a single reagent/test lot. Details are presented in Table 1.

### Seroconversion panels

Consisted of a total of 60 plasma samples, collected in the USA, from five differ-ent SARS-CoV-2 NAT positive individuals at regular intervals from early infection to approximately 8 weeks post-symptoms. Results of testing were used to determine the number of days post-onset of symptoms the test kit first detected reactivity.

### Sero-reversion panels

Consisted of 47 plasma samples obtained from 10 individuals in Germany, starting from no less than 18 days to no greater than 50 days post-infection. Each individual had between three and six bleeds each. The results were used to determine the ability of the test kit to detect waning IgM-specific antibodies.

### Repeatability

To determine the repeatability of each test kit, a commercial quality control sample (DiaMex, Heidelberg Germany) was tested 30 times in the same test run. The %CV was calculated.

### Testing and reporting protocol

NRL aliquoted the panel samples into single-use vials and randomized the vials into a complete evaluation panel, which was stored at −20°C until use. Only NRL knew the code for the randomization. The complete, randomized panel of samples was provided to the

testing laboratory or the test kit manufacturer for testing. The complete panels were shipped on dry ice and stored frozen at the testing laboratory until use. All testing was performed as per the manufacturer's instructions for use (IFU). Results were provided to NRL as a Microsoft Excel file and/or as a copy of the printed results sheet. The results were copied or transcribed into further Microsoft Excel spreadsheets for decoding and analysis by NRL staff. All manual and electronic transcriptions were cross-checked by a second person.

## ACKNOWLEDGMENTS

We thank NRL scientific staff: Sadaf Mohiuddin, Jingjing Cai, Bethmi Liyanage, and all technical support staff who took part in composing the panels and performing testing. In addition, we acknowledge GreenLight Clinical Laboratories, Douglass Handly Moir Pathology, and the Austin Pathology Laboratories for testing panel samples on respective platforms. In particular, we acknowledge Technopath Clinical Diagnostics (Ballina, Ireland) for their generous donation of positive donor samples including the seroconversion samples. Other panel samples were obtained commercially from Boca Biolisitics (Fl, USA); BioMex (Heidelberg, Germany); Medical Research Networx (Fl, USA); and Seracare (MA, USA).

## AUTHOR AFFILIATION

[1]National Serology Reference Laboratory, Fitzroy, Victoria, Australia

## AUTHOR ORCIDs

Wayne Dimech  http://orcid.org/0000-0003-3425-9419

## AUTHOR CONTRIBUTIONS

Wayne Dimech, Conceptualization, Data curation, Formal analysis, Methodology, Supervision, Validation, Visualization, Writing – original draft | Shannon Curley, Data curation, Formal analysis, Methodology, Supervision, Validation, Writing – review and editing | Jing Jing Cai, Formal analysis, Investigation, Resources, Supervision, Validation, Writing – review and editing

## ADDITIONAL FILES

The following material is available online.

### Supplemental Material

**Fig. S1 (Spectrum03228-23-s0001.xlsx).** Results of testing five seroconversion panels on 25 SARS-CoV-2 automated antibody tests.
**Fig. S2 (Spectrum03228-23-s0002.xlsx).** Results of testing 10 sero-reversion panels on nine SARS-CoV-2 IgM-specific automated test kits.

### Open Peer Review

**PEER REVIEW HISTORY (review-history.pdf).** An accounting of the reviewer comments and feedback.

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

## AUTHOR BIO

Wayne Dimech BAppSc, MASM, FAIMS, MBA, FFSc (RCPA), is Executive Manager, Scientific and Business Relations at NRL in Melbourne. He is a recognized expert in the control and standardization of infectious disease serology and laboratory quality systems.

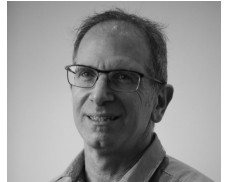

