## [Reviewer comments · Microbiology Spectrum]

Microbiology Spectrum

Comprehensive, Comparative Evaluation of 25 Automated SARS-CoV-2 Serology Assays

Wayne Dimech, Shannon Curley, and JingJing Cai

Corresponding Author(s): Wayne Dimech, National Reference Laboratory, Australia

Review Timeline:

Submission Date:	August 31, 2023
Editorial Decision:	October 3, 2023
Revision Received:	October 13, 2023
Accepted:	October 26, 2023

Editor: Oliver Laeyendecker

Reviewer(s): Disclosure of reviewer identity is with reference to reviewer comments included in decision letter(s). The following individuals involved in review of your submission have agreed to reveal their identity: Adolfo Firpo-Betancourt (Reviewer #2)

Transaction Report:

DOI: <https://doi.org/10.1128/spectrum.03228-23>

October 3, 2023

Dr. Wayne Dimech
National Reference Laboratory, Australia
4th Floor Healy Building
41 Victoria Parade
Fitzroy, Victoria 3065
Australia

Re: Spectrum03228-23 (Comprehensive, Comparative Evaluation of 25 Automated SARS-CoV-2 Serology Assays)

Dear Dr. Wayne Dimech:

Link Not Available

Sincerely,

Oliver Laeyendecker

Journals Department
Reviewer comments:

Reviewer #1 (Comments for the Author):

Summary

This manuscript discusses the evaluation of COVID-19 serology test kits. Initially, rapid serology tests emerged following the recognition of the novel coronavirus (SARS-CoV-2) in November 2019, with limited regulatory oversight. The National Serology Reference Laboratory (NRL) in Australia conducted a comprehensive study to assess the performance of 16 IgG or Total antibody test kits.

Key findings from the evaluation include: Most test kits for detecting IgG or total antibodies exhibited good concordance with recent infection, with the Ortho VITROS Total kit performing exceptionally well. Test kits generally demonstrated high clinical

specificity, although there were slight variations, with some kits reporting lower specificity. Variability in analytical sensitivity was observed between test kits and sample dilutions, but no significant differences were found between different lot numbers of the same test kit. Some test kits showed cross-reactivity with certain substances, but most had low rates of false reactivity. The manuscript highlights the importance of rigorous evaluation protocols, especially during health emergencies, to ensure the reliability of diagnostic tests. It also emphasizes the need for improved regulatory standards and a network of expert laboratories to respond swiftly to future outbreaks.

Major Concerns

- Overall, the methods are vague. It seems there was likely an intention to cite previous work to save space in the manuscript but I don't see any citations in the methods.
 - o Study protocol, sample info (no citation), panels
- While the overall message that improved regulatory standards are needed as tests were created quickly, perhaps haphazardly, and pushed out under emergency use authorization remains the same, it is notable that the number of samples and the limited patient population largely impact the generalizability of the study. Given that in addition to the higher performing tests in this manuscript compared to the previous 35 assays studied, I would urge the authors to consider adding a reasonable amount of samples throughout the pandemic's variant waves.

Minor Concerns

- Limited patient population. 199 commercially acquired from non-hospitalized patients from the USA or Germany.
- Limited diversity of viral strains in samples due to limited timeframe of Jan to April of 2020. This lowers the generalizability of the results to an early stage of the pandemic.
- The discussion section should include a paragraph about study limitations that should include the two bullets above as well as others readers may not know from reading the manuscript.

Reviewer #2 (Comments for the Author):

Thank you for this excellent overview of major automated serological testing methods for relevant antibodies to SARS-CoV-2. It provides a useful and meaningful framework for assessing local real-life clinical laboratory medical practice experience employing any of these methods and instrument systems. It really puts in perspective the challenges of serological testing for infectious diseases which you have pointed out so clearly in your review published in 2021. It helps to answer frequent concerns with serological test results raised by clinicians and epidemiologists.

Specific comments on the content:

Check line 134 for possible spelling error: RTD for RDT?

Lines: 92-99 Cross-reacting and interfering substances.

I miss the specific analytical methods of assays with cross reactive results and the identification of the specific interfering substances affecting the assays mentioned in your discussion. Adding these to the tables would add value to clinical laboratory medicine practitioners.

Lines 119-120: "... the utility of these tests was unknown at the time.." There was no consensus on the proper utilization of the serological tests at a time when the real need was for a reliable diagnostic test. Promoting serological testing then was difficult because of the potential false negatives during early infection which would have been disastrous. It was clear from the start that seroprevalence would have been useful. If testing had been done earlier human to human transmission would have been recognized earlier. Another point on medical utility of serological testing early during a new epidemic is the selection of high immune responders (plasma with high antibody titers) as potential donors of convalescent plasma for treatment of severe COVID-19 patients. Another major concern then was the cost of serological testing as pointed out in your first reference (1). The potential benefits for serological testing are well described in your reference 19.

Staff Comments:

Preparing Revision Guidelines

Please return the manuscript within 60 days; if you cannot complete the modification within this time period, please contact me. If you do not wish to modify the manuscript and prefer to submit it to another journal, please notify me of your decision immediately so that the manuscript may be formally withdrawn from consideration by Microbiology Spectrum.

Dear Editor,

Thank you for the opportunity to respond to the reviewers' comments. Please find below a point-by-point response for your consideration. An updated manuscript and clean PDF have been submitted as indicated.

Reviewer comments:

Reviewer #1 (Comments for the Author):

Summary

This manuscript discusses the evaluation of COVID-19 serology test kits. Initially, rapid serology tests emerged following the recognition of the novel coronavirus (SARS-CoV-2) in November 2019, with limited regulatory oversight. The National Serology Reference Laboratory (NRL) in Australia conducted a comprehensive study to assess the performance of 16 IgG or Total antibody test kits. Key findings from the evaluation include: Most test kits for detecting IgG or total antibodies exhibited good concordance with recent infection, with the Ortho VITROS Total kit performing exceptionally well. Test kits generally demonstrated high clinical specificity, although there were slight variations, with some kits reporting lower specificity. Variability in analytical sensitivity was observed between test kits and sample dilutions, but no significant differences were found between different lot numbers of the same test kit. Some test kits showed cross-reactivity with certain substances, but most had low rates of false reactivity.

The manuscript highlights the importance of rigorous evaluation protocols, especially during health emergencies, to ensure the reliability of diagnostic tests. It also emphasizes the need for improved regulatory standards and a network of expert laboratories to respond swiftly to future outbreaks.

Major Concerns

- Overall, the methods are vague. It seems there was likely an intention to cite previous work to save space in the manuscript but I don't see any citations in the methods.

Author - *Indeed Reviewer 1 is correct in that the intention was to cite previous study also published in Spectrum (Reference 17). It was determined that a detailed repetition of the panel design was not required. However, on re-reading the submitted version of this publication, the previous study, while referenced, was not sufficiently clear.*

To rectify this, we have added reference 17 to the first sentence of "Sample panels" section of "Methods", which reads "The panels of samples used in the study is presented in detail elsewhere".

*In addition, in the second sentence of paragraph two of the "Introduction", we have added extra information highlighted in bold "In addition to this WHO study, NRL offered a similar evaluation service to other manufacturers of laboratory-based, automated SARS-CoV-2 serology tests, **drawing from the same panels of samples.**"*

o Study protocol, sample info (no citation), panels

Author - *As above*

- While the overall message that improved regulatory standards are needed as tests were created quickly, perhaps haphazardly, and pushed out under emergency use authorization remains the same, it is notable that the number of samples and the limited patient population largely impact the generalizability of the study. Given that in addition to the higher performing tests in this manuscript compared to the previous 35 assays studied, I would urge the authors to consider adding a reasonable amount of samples throughout the pandemic's variant waves.

Author – *The study was conducted at the time of the emergence of the pandemic and represents one of the most comprehensive such studies published, especially when combined with the previously reported rapid test evaluation. Although the Reviewer's comments are valid, as this study included positive samples containing antibodies derived only from individuals infected with the original Wuhan Strain and does not consider the performance of these test kits to detect antibodies derived from an evaluation of new VoC, or from different vaccines or vaccination programs. However, we would argue that this study acts a bench-mark. It may be useful to have future similar studies focusing on different origins of antibodies, but that was out of scope for this study, and was not available at the time due to the stage of the pandemic.*

Minor Concerns

- Limited patient population. 199 commercially acquired from non-hospitalized patients from the USA or Germany.

Author – *As discussed above, this study serves as a bench-mark. Its aim was to compare the performance of test kits using the same panel of samples. Access to high volume samples during a pandemic was challenging and costly. Ideally, samples from many countries may have been advantageous. The number of samples (n=199), however is not problematic as this number of samples allows for adequate confidence to assess sensitivity. Notwithstanding, we have added a sentence **"A limitation of this study is that it used commercially acquired from non-hospitalized patients from the USA or Germany in the panel of positive samples, acquired early in the pandemic. Therefore, the SARS-CoV-2 antibodies detected were post infection with the Wuhan strain. The ability of these assay to detect antibodies arising from infections with other variants of concern or post immunisation was not assessed"** at the end of the second last paragraph of the discussion.*

- Limited diversity of viral strains in samples due to limited timeframe of Jan to April of 2020. This lowers the generalizability of the results to an early stage of the pandemic.

Author- *addressed above*

- The discussion section should include a paragraph about study limitations that should include the two bullets above as well as others readers may not know from reading the manuscript.

Author- *addressed above*

Reviewer #2 (Comments for the Author):

Thank you for this excellent overview of major automated serological testing methods for relevant antibodies to SARS-CoV-2. It provides a useful and meaningful framework for assessing local real-life clinical laboratory medical practice experience employing any of these methods and instrument systems. It really puts in perspective the challenges of serological testing for infectious diseases which you have pointed out so clearly in your review published in 2021. It helps to answer frequent concerns with serological test results raised by clinicians and epidemiologists.

Specific comments on the content:

Check line 134 for possible spelling error: RTD for RDT?

Author - corrected

Lines: 92-99 Cross-reacting and interfering substances.

I miss the specific analytical methods of assays with cross reactive results and the identification of the specific interfering substances affecting the assays mentioned in your discussion. Adding these to the tables would add value to clinical laboratory medicine practitioners.

Author – *The specific cross reacting and interfering substances that were mentioned in the results section are added under “Cross-reacting and interfering substances” in the “Results” section. The tests mentioned in the discussion were from the other, sister study previously published.*

Lines 119-120: "... the utility of these tests was unknown at the time.." There was no consensus on the proper utilization of the serological tests at a time when the real need was for a reliable diagnostic test. Promoting serological testing then was difficult because of the potential false negatives during early infection which would have been disastrous. It was clear from the start that seroprevalence would have been useful. If testing had been done earlier human to human transmission would have been recognized earlier. Another point on medical utility of serological testing early during a new epidemic is the selection of high immune responders (plasma with high antibody titers) as potential donors of convalescent plasma for treatment of severe COVID-19 patients. Another major concern then was the cost of serological testing as pointed out in your first reference (1). The potential benefits for serological testing are well described in your reference 19.

Author – *We agree with the comments of Reviewer 2. In retrospect, seroprevalence studies and the possibility of identifying high immune responder samples were possible uses for serology. The comment was meant to indicate the “diagnostic” utility of antibody testing was unknown. I have added “**diagnostic utility**” to clarify.*

Re: Spectrum03228-23R1 (Comprehensive, Comparative Evaluation of 25 Automated SARS-CoV-2 Serology Assays)

Dear Dr. Wayne Dimech:

Your manuscript has been accepted, and I am forwarding it to the ASM production staff for publication. Your paper will first be checked to make sure all elements meet the technical requirements. ASM staff will contact you if anything needs to be revised before copyediting and production can begin. Otherwise, you will be notified when your proofs are ready to be viewed.

Sincerely,
Oliver Laeyendecker
Editor
Microbiology Spectrum